# In Vitro Interactions of TiO_2_ Nanoparticles with Earthworm Coelomocytes: Immunotoxicity Assessment

**DOI:** 10.3390/nano11010250

**Published:** 2021-01-19

**Authors:** Natividad Isabel Navarro Pacheco, Radka Roubalova, Jaroslav Semerad, Alena Grasserova, Oldrich Benada, Olga Kofronova, Tomas Cajthaml, Jiri Dvorak, Martin Bilej, Petra Prochazkova

**Affiliations:** 1Institute of Microbiology of the Czech Academy of Sciences, Videnska 1083, 142 20 Prague 4, Czech Republic; natividad.pacheco@biomed.cas.cz (N.I.N.P.); r.roubalova@biomed.cas.cz (R.R.); jaroslav.semerad@biomed.cas.cz (J.S.); alena.grasserova@biomed.cas.cz (A.G.); benada@biomed.cas.cz (O.B.); kofra@biomed.cas.cz (O.K.); cajthaml@biomed.cas.cz (T.C.); dvorak@biomed.cas.cz (J.D.); mbilej@biomed.cas.cz (M.B.); 2First Faculty of Medicine, Charles University, Katerinska 1660/32, 121 08 Prague 2, Czech Republic; 3Faculty of Science, Institute for Environmental Studies, Charles University, Benatska 2, 128 01 Prague 2, Czech Republic

**Keywords:** earthworm, coelomocyte, TiO_2_ nanoparticles, reactive oxygen species, innate immunity, lipid peroxidation, alkaline comet assay, phagocytosis, apoptosis, gene expression

## Abstract

Titanium dioxide nanoparticles (TiO_2_ NPs) are manufactured worldwide. Once they arrive in the soil environment, they can endanger living organisms. Hence, monitoring and assessing the effects of these nanoparticles is required. We focus on the *Eisenia andrei* earthworm immune cells exposed to sublethal concentrations of TiO_2_ NPs (1, 10, and 100 µg/mL) for 2, 6, and 24 h. TiO_2_ NPs at all concentrations did not affect cell viability. Further, TiO_2_ NPs did not cause changes in reactive oxygen species (ROS) production, malondialdehyde (MDA) production, and phagocytic activity. Similarly, they did not elicit DNA damage. Overall, we did not detect any toxic effects of TiO_2_ NPs at the cellular level. At the gene expression level, slight changes were detected. Metallothionein, fetidin/lysenin, lumbricin and MEK kinase I were upregulated in coelomocytes after exposure to 10 µg/mL TiO_2_ NPs for 6 h. Antioxidant enzyme expression was similar in exposed and control cells. TiO_2_ NPs were detected on coelomocyte membranes. However, our results do not show any strong effects of these nanoparticles on coelomocytes at both the cellular and molecular levels.

## 1. Introduction

Titanium dioxide nanoparticles (TiO_2_ NPs) are commonly used in different industries because of their physico-chemical properties. TiO_2_ NPs have photocatalytic properties, protect against UV radiation, are used as semiconductors, etc. These nanoparticles are used, e.g., in cosmetics, food industry, paints, ceramics, devices development, and the agriculture industry [1,2,3]. In the last decade, TiO_2_ NPs have been used in wastewater treatment plants for their ability to degrade some organic pollutants [1]. Thus, TiO_2_ NPs reach the soil system from different sources including sludge, nanofertilizers, and nanopesticides. These nanoparticles then interact with the soil biota. It is therefore very important to assess the potential risk of TiO_2_ NPs to soil organisms.

Earthworms are dominant soil invertebrate animals. They possess a strong immune system because of their permanent contact with soil bacteria, viruses, and fungi. Defense mechanisms are used in earthworm protection against soil pollutants including nanoparticles. Earthworms *Eisenia andrei* and *E. fetida* are used as model organisms to monitor ecotoxicity according to OECD guidelines [4,5,6]. TiO_2_ NPs do not affect earthworm viability and growth [2,3]. In some cases, reproductive inhibition was observed [7]. Further, these nanoparticles can induce, e.g., oxidative stress, DNA damage, apoptosis, and affect gene expression [3]. Earthworm cellular defense mechanisms are based on coelomocytes present in the coelomic fluid. Coelomocytes can be divided into free chloragogen cells called eleocytes, with a mainly nutritive function, and amoebocytes, which are the immune effector cells [8]. Amoebocytes can be further divided into granular (GA) and hyaline (HA) amoebocytes.

Various nanoparticles were described to impair earthworm defense mechanisms. Hayashi et al. showed that Ag NPs altered the expression of some genes involved in coelomocyte oxidative stress and immune reactions [9]. Further, Ag nanowires detected on coelomocyte membranes increased intracellular esterase activity [10]. ZnO NPs were internalized by coelomocytes, with consequent DNA damage [11]. However, similar mechanisms were not described for TiO_2_ NPs in earthworms. TiO_2_ NPs cause significant mitochondrial dysfunction by increasing mitochondrial ROS levels and decreasing ATP generation in macrophages. Moreover, TiO_2_ NPs exposure activated inflammatory responses and attenuated macrophage phagocytic function [12]. TiO_2_ NPs interacted with sea urchin immune cells and increased the antioxidant metabolic pathway in vitro [13]. In earthworms, only increased apoptosis was observed following TiO_2_ nanocomposites exposure [7,14,15,16].

A compromised immune system may result in a decreased reproductive rate and increased mortality of earthworms. Thus, nanoparticle toxicity risk assessment is extremely important, as the adverse health effects remain poorly characterized for many nanomaterials. We aimed to assess the potentially dangerous impact of TiO_2_ NPs exposure on earthworms’ cellular function, including the immune responses to harmful stimuli.

*E. andrei* coelomocytes were exposed to 1, 10, and 100 µg/mL of TiO_2_ NPs for 2, 6, and 24 h in vitro. After exposure, viability, oxidative stress (reactive oxygen species and malondialdehyde production), immune functions (phagocytosis), and genotoxicity (DNA damage) were assessed. Further, electron microscopy (transmission and scanning) enabled TiO_2_ NPs localization on the cell surface. Gene expression changes were also followed to better understand the underlying cellular mechanisms.

## 2. Materials and Methods

### 2.1. Animal Handling, Sample Collection, and Culture Medium Preparation

Clitelate, adult *Eisenia andrei* earthworms were obtained from the laboratory compost breeding. Earthworms were first kept on moist filter paper for 48 h to depurate their guts. Coelomocytes were harvested by applying 2 mL of extrusion buffer (5.37 mM EDTA (Sigma-Aldrich, Steinheim, Germany); 50.4 mM guaiacol glyceryl ether (GGE; Sigma-Aldrich, Steinheim, Germany) in Lumbricus Balanced Salt Solution (LBSS; [17]) per earthworm for 2 min. The cells were then centrifuged and washed twice in LBSS (200× *g*, 4 °C, 10 min). Subsequently, cells were counted and diluted to 10^6^ cells/well for scanning electron microscopy (SEM) and lipid peroxidation assessment. 1 × 10^5^ cells/well, 2 × 10^5^ cells/well, and 3 × 10^5^ cells/well were used for the ROS production analysis, apoptosis detection, and phagocytosis assay, respectively.

RPMI 1640 culture medium supplemented with 5% heat-inactivated fetal bovine serum (FBS; Life technologies, Carlsbad, USA), 1 M HEPES (4-(2-hydroxyethyl)-1-piperazineethanesulfonic acid; pH 7.0–7.6, Sigma-Aldrich; Gillingham, UK), 100 mM sodium pyruvate (Sigma-Aldrich, Steinheim, Germany), 100 mg/mL gentamycin (Corning, Manassas, VA, USA), and antibiotic–antimycotic solution (Sigma-Aldrich, Steinheim, Germany) was diluted with autoclaved MilliQ-water to 60% (*v*/*v*) to obtain R-RPMI 1640 medium [18]. Subsequently, TiO_2_ NPs were dispersed in R-RPMI 1640 medium and incubated with cells in darkness at 20 °C for 2, 6, and 24 h in triplicate. 

### 2.2. TiO_2_ NPs Characterization

Aeroxide TiO_2_ P25 nanoparticles (irregular and semi-spherical shape; mexoporous NPs, anatase, and rutile 4:1; primary size between 10 and 65 nm) were purchased from Evonik Degussa (Essen, Germany). TiO_2_ NPs were previously characterized in several aqueous solutions, as described by Brunelli et al. [19]. Nanoparticle physico-chemical properties were determined by ZetaSizer Ultra (Panalytical Malvern; Malvern, UK), transmission electron microscope (TEM), and TECAN 200 Pro plate reader. Powder TiO_2_ NPs were weighed and dispersed in distilled water. Then, diluted TiO_2_ NPs were vortexed thoroughly for 5 min prior to further dilution [20]. TiO_2_ NPs were diluted either in R-RPMI 1640 medium or distilled water to a concentration of 1, 10, and 100 µg/mL, and incubated for 2, 6, and 24 h. Experiments were carried out in triplicate. Culture medium and distilled water without NPs were used as negative controls.

### 2.3. Electron Microscopy Analyses

#### 2.3.1. Cell Preparation

Coelomocytes were exposed to 1, 10, and 100 µg/mL TiO_2_ NPs for 2, 6, and 24 h. Cell viability was measured by propidium iodide (PI; 1 µg/mL) staining using flow cytometer. Then, samples were collected and fixation solution (5% glutaraldehyde in PBS) was added in a 1:1 ratio (*v*:*v*). Fixed cells were shaken gently for 15 min and kept overnight at 4 °C. 

#### 2.3.2. Scanning Electron Microscopy (SEM)

For SEM, fixed cells were washed with LBSS buffer three times at room temperature for 20 min, and centrifuged at 150× *g* for 10 min. Then, they were allowed to adhere onto poly-L-lysine coated round 13 mm Thermanox Plastic Coverslips (Nunc, Thermo Fisher Scientific; Roskilde, Denmark) overnight at 4 °C. The coverslips with attached cells were washed with ddH_2_O and fixed with 1% OsO_4_ for one hour at room temperature. The coverslips were then washed three times for 20 min, dehydrated through an alcohol series (25, 50, 75, 90, 96, and 100%), and were critical-point dried from liquid CO_2_ in a K850 Critical Point Dryer (Quorum Technologies Ltd., Ringmer, UK). The dried coverslips were sputter-coated using a high-resolution Turbo-Pumped Sputter Coater Q150T (Quorum Technologies Ltd., Ringmer, UK) with 3 nm of platinum. Alternatively, for EDS microanalysis, the samples were coated with 10 nm of silver or 5 nm of carbon. The final samples were examined in a FEI Nova NanoSEM scanning electron microscope (FEI, Brno, Czech Republic) at 5 kV using CBS and TLD detectors. An electron beam deceleration [21] mode of the Nova NanoSEM scanning electron microscope performed at a StageBias of 883.845 V and accelerating voltage of 5 kV was used for high-resolution imaging. The EDS microanalysis was performed at 15 kV using an Ametek^®^ EDAX Octane Plus SDD detector and TEAM™ EDS Analysis Systems (AMETEK B. V.; Tilburg, The Netherlands).

#### 2.3.3. Transmission Electron Microscopy (TEM)

For TEM, a TiO_2_ NPs suspension (500 µg/mL; 5 µL) was applied onto glow-discharge-activated [22] carbon-coated 400-mesh copper grids (G400, SPI Supplies, Structure Probe, Inc., West Chester, PA, USA). Nanoparticles were sedimented for 1 min and the remaining solution was then blotted with filter paper and the grids were air-dried. A Philips CM100 electron microscope (Philips EO, Eindhoven, The Netherlands; Thermo Fisher Scientific) equipped with a Veleta slow-scan CCD camera (EMSIS GmbH, Muenster, Germany) was used to examine the grids. TEM images were processed in the proprietary iTEM software (EMSIS GmbH, Muenster, Germany).

### 2.4. Flow Cytometry Assays

Coelomocytes were incubated with TiO_2_ NPs (1, 10, and 100 µg/mL) for 2, 6, and 24 h. Cells were then treated as described below and analyzed with a laser scanning flow cytometer. Through flow cytometry, coelomocytes were subdivided into eleocytes, granular (GA), and hyaline amoebocytes (HA). The coelomocytes subset detection was based on the cell size (FSC) and the cell inner complexity/granularity (SSC). Cell viability was assessed for every assay. All flow cytometry assays were performed by three independent experiments with three replicates per each treatment and time interval. The minimum collected events were 1000 per population. Event counts per each gate were calculated by Flowjo (9.9.4 version, BD Biosciences, San Jose, CA, USA). In each flow cytometry assay, coelomocytes were exposed to H_2_O_2_ as a positive control (Sigma-Aldrich, Steinheim, Germany; 10 mM H_2_O_2_ for 30 min incubation for apoptosis and phagocytosis, and 1 mM H_2_O_2_ for ROS production assesment). Controls with and without PI (1 mg/L; Sigma-Aldrich, Steinheim, Germany) were included in each experiment. Further, control analysis of 1, 10, and 100 µg/mL TiO_2_ NPs incubated with or without cells for 2, 6, and 24 h were performed (Appendix A).

For ROS production determination, 20.6 µM 2′,7′-dichlorofluorescin diacetate (DCF-DA; Sigma-Aldrich, Steinheim, Germany) was added to the washed cell suspension (LBSS, 200× *g*, 4 °C, 10 min) for 15 min in darkness. Subsequently, the cell suspension was washed twice with LBSS (200× *g*, 4 °C, 10 min) and stained with PI. 

To detect the apoptotic process a cell suspension was washed twice with Annexin V buffer (200× *g*, 4 °C, 10 min; 0.01 M HEPES (pH 7.4), 0.14 M NaCl, and 2.5 mM CaCl_2_ solution), and subsequently stained with 5 µL of Alexa Fluor 647-Annexin V (15 min in darkness; Thermo Fisher Scientific, Eugene, OR, USA). PI was then added to the cell suspension and measured by flow cytometry. The apoptosis % represented the apoptotic cell number out of each subpopulation. The necrosis % represented the necrotic cell number out of each subpopulation.

The phagocytosis assay was performed using latex beads (Fluoresbrite^®^ Plain YG; 1 µm microspheres diameter; Polysciencies Inc., Warrington, PA, USA) added to the incubation plates in a 1:100 ratio (cells:beads) and kept in darkness at 17 °C for 18 h. Then, cell suspensions were washed twice with LBSS (200× *g*, 4 °C, 10 min), stained with PI, and analyzed by flow cytometry. The % phagocytic activity was determined by the % of alive cells, which were able to engulf at least one bead out of each subpopulation. Each experiment included samples with NPs dispersed in the medium in order to detect effects exerted by NPs alone. 

### 2.5. Malondialdehyde (MDA) Production and Alkaline Comet Assay

Coelomocytes were incubated with TiO_2_ NPs (10 and 100 µg/mL) or CuSO_4_ (100 µg/mL; positive control) for 2, 6, and 24 h. Afterward, cell suspensions were collected and MDA production was measured. MDA production was detected by high-performance liquid chromatography with fluorescence detection (HPLC/FLD) using derivatized MDA-TBA2 [23]. MDA analysis was performed in three independent experiments with 3 replicates for each treatment and time interval.

For the alkaline comet assay, 1.5 × 10^4^ cells exposed to 1, 10, and 100 µg/mL TiO_2_ NPs for 2, 6, and 24 h were mixed with 2% 2-hydroxyethyl agarose (Sigma-Aldrich, Steinheim, Germany) at 37 °C. Glass slides containing agarose with cells were kept at 4 °C for 10 min. Subsequently, samples were incubated for 2 h in lysis buffer (2.5 M NaCl, 10 mM Tris-HCl, 100 mM EDTA, 1% Triton X-100, pH 10). Then, slides were immersed three times in unwinding buffer (0.03 M NaOH, 2 mM EDTA, pH 12.7) for 20 min. Gel electrophoresis was carried out at 24 V, 300 mA for 25 min. Subsequently, slides were rinsed with neutralizing buffer (0.4 M Tris, pH 7.5) and stained with PI (3 µg/mL) for 20 min. The excess dye was removed with distilled water (5 min). Then, samples were stored in humidified chambers until the analysis by LUCIA Comet Assay software. One hundred cells per replicate of each treatment and time interval were analyzed, and the mean of DNA content in 100 comet tails (%) was calculated as a parameter of DNA damage. Positive control (100 mM H_2_O_2_; 30 min incubation; Sigma-Aldrich, Steinheim, Germany) was included with the assay. The comet assay was repeated in three independent experiments with three replicates for each treatment and time interval.

### 2.6. mRNA Levels Quantification 

Cells were incubated with TiO_2_ NPs (1 and 10 µg/mL) for 2, 6, and 24 h. Cellular RNA was isolated using the RNAqueous^®^-Micro Kit (Invitrogen, Vilnius, Lithuania). RNA (500 ng) was reverse-transcribed with the Oligo(dT)12–18 primer and Superscript IV Reverse Transcriptase (Life Technologies). Non-RT controls were included to show the elimination of gDNA contamination.

Quantitative PCR (CFX96 Touch™ Real-Time PCR detection System, Bio-Rad) was performed to detect changes in mRNA levels encoding proteins participating in metal detoxification (metallothionein, phytochelatin), oxidative stress (manganese superoxide dismutase, Mn-SOD; copper-zinc-superoxide dismutase CuZn-SOD; catalase), immunity (endothelial monocyte-activating polypeptide II, EMAPII; fetidin/lysenin, and lumbricin), and signal transduction (MEK kinase I, MEKK I; and protein kinase C I, PKC I). Sequences of primers used in qPCR assays are referred in Appendix A. The PCR reactions were performed in a 25 µL volume containing 4 µL of cDNA (dilution 1:10, except for 1:5 dilution for SODs). The cycling parameters were similar to Roubalova et al., with slight changes [24]: 4 min at 94 °C, 35 cycles of 10 s at 94 °C, 25 s at 60 °C (at 58 °C for MEKK I, PKC I, and catalase), 35 s at 72 °C, and a final extension for 7 min at 72 °C. Gene expression changes were calculated according to the 2−ΔΔCT (Livak) method. Two reference genes (RPL13, RPL17) were selected as internal controls for gene expression normalization. Non-template control was included in each experiment. The fold change in the mRNA level was related to the change of the corresponding controls. The results were expressed as the mean ± SEM of the values. mRNA levels quantification was performed by three independent experiments with duplicates per each treatment and time interval.

### 2.7. Statistical Analyses

Statistical analyses were performed using GraphPad Prism (8.3.1 version, San Diego, CA, USA). Flow cytometry assays, lipid peroxidation, alkaline comet assay, and gene expression were analyzed by two-way ANOVA with Bonferroni post-test.

## 3. Results

### 3.1. TiO_2_ NPs Characterization

TiO_2_ NPs were dispersed and stabilized in distilled water and in R-RPMI 1640 culture medium evenly. However, differences in NPs characteristics were observed between both mediums along the exposure time (2, 6, and 24 h) (Table 1). In the UV/Vis spectra, NPs exerted a similar wavelength range: 300–370 nm for distilled water; 320–380 nm for R-RPMI 1640 medium. Although NPs absorbed similar UV/Vis wavelengths, differences were observed in the hydrodynamic size distribution. TiO_2_ NPs dispersed in R-RPMI 1640 medium were not stabilized and tended to aggregate. The aggregation was detected between 6 and 24 h of incubation. After 6 h, the hydrodynamic size distribution was 35.5 ± 3.94 nm, and it increased to 597 ± 447 nm after 24 h. At 2–6 h, the hydrodynamic size of TiO_2_ NPs was stable (31.34 ± 1.55 to 35.5 ± 3.94 nm, respectively). In distilled water, the hydrodynamic size distribution was stable between 2–24 h 581 ± 23.30 and 480 ± 64.3 nm, respectively. Regarding zeta potential, TiO_2_ NPs dispersed in both distilled water and R-RPMI 1640 did not change significantly over time (Table 1). 

### 3.2. Electron Microscopy

The TiO_2_ NPs size given by the manufacturer was 10–65 nm. However, we were not able to verify this information because of a great aggregation of TiO_2_ NPs in concentrations detectable by TEM. According to our measurements, the nanoparticles ranged between 20 and 100 nm. The TiO_2_ NPs were rode/spherical (Figure 1).

In coelomocytes exposed to 100 µg/mL TiO_2_ NPs, nanoparticles were observed on cell membranes by scanning electron microscopy (Figure 2). Moreover, EDS microanalysis confirmed Ti presence in nanoparticle clusters on the cell surface (Figure 3). At 10 µg/mL TiO_2_ NPs exposure, nanoparticles were also detected on the coelomocyte surface but with lower frequency. EDS microanalysis of non-treated cells is shown in Appendix A.

### 3.3. Flow Cytometry Assays

Coelomocyte subpopulations were differentiated by flow cytometry (Appendix A). Thus, the viability of HA and GA were analyzed. The eleocyte subpopulation was excluded from the results because of the interaction between their autofluorescence and the fluorescences used in the assays.

HA and GA viability (the percentage of alive cells in each subpopulation) was similar in non-treated cells and TiO_2_ NPs-exposed cells. No differences in viability were observed between amoebocyte subpopulations. 

We did not observe any significant changes in ROS production in HA or in GA after exposure to any of the TiO_2_ NPs concentrations (Figure 4). HA population exerted two times lesser fluorescence intensity in comparison to the GA population. This suggests that HA population is less potent to produce ROS than GA population (Figure 4). Illustrative histograms of ROS production between control samples and positive control (1 mM H_2_O_2_), indicating a clear shift in sample fluorescence, are shown in Appendix A.

Similarly, we did not detect any significant differences in the apoptosis level between TiO_2_ NPs exposed and control cells (both in HA and GA; Figure 5 and Figure 6). In both populations, the early apoptosis percent is similar over time, while late apoptosis slightly decreased after 24 h (Figure 5 and Figure 6). Necrosis increased along the exposure time in GA (Figure 6). However, statistically significant differences were not detected between treated and control cells. Representative distributions of the apoptotic/necrotic cell stages in GA and HA cell subpopulations are shown in Appendix A, respectively.

The viable amoebocyte phagocytic activity was measured in both amoebocyte subsets (HA and GA). Representative phagocytic activity density plots of GA and HA cell subpopulations are shown in Appendix A, respectively. The phagocytic activity was similar in both amoebocyte subpopulations (GA and HA; Figure 7). A decrease in the phagocytic activity of HA control cells and TiO_2_ NPs-exposed cells occurred after 24 h (Figure 7). This slight decrease may indicate the greater sensitivity of HA to external conditions. However, phagocytic activity was not significantly affected by NPs treatment or by the incubation time. Phagocytic activity of untreated cells with and without Fluoresbrite^®^ YG Plain 1µm microspheres was also compared (Appendix A).

### 3.4. MDA and Alkaline Comet Assay

Malondialdehyde (MDA) is a lipid peroxidation subproduct, and it is therefore used as an oxidative stress biomarker in cells. We did not detect any significant increase in MDA production in cells exposed to TiO_2_ NPs (10 and 100 µg/mL) at the tested timepoints (2, 6, and 24 h; Figure 8). 

The DNA damage in coelomocytes exposed to 1, 10, and 100 µg/mL TiO_2_ NPs for 2, 6, and 24 h was assessed by the alkaline comet assay. DNA damage was evaluated by the mean tail intensity (% DNA in tail) of 100 comets in each incubation. The observed DNA damage was not greater than 40% during exposure with TiO_2_ NPs and the non-treated cells (Figure 9).

### 3.5. mRNA Levels of Detoxification, Immune, Antioxidant, and Signal Transduction Molecules

The change in mRNA levels of appropriate molecules after coelomocyte exposure to TiO_2_ NPs was assessed (Table 2). Metallothioneins involved in metal detoxification were significantly upregulated in coelomocytes exposed to 1 µg/mL TiO_2_ NPs for 2, 6 and 24 h, and in coelomocytes exposed to 10 µg/mL TiO_2_ NPs for 6 h. Further, significant Mn-SOD downregulation was detected in coelomocytes incubated with 10 µg/mL TiO_2_ NPs for 6 h. Then, fetidin/lysenin and lumbricin were upregulated upon coelomocyte exposure to 10 µg/mL TiO_2_ NPs for 6 h. MEKK I upregulation after 1 µg/mL TiO_2_ NPs exposure for 24 h, and PKC I downregulation after 10 µg/mL TiO_2_ NPs exposure for 6 and 24 h were detected. Surprisingly, the mRNA levels of catalase and CuZn-SOD (antioxidant enzymes) were not significantly altered. 

## 4. Discussion

The physico-chemical properties of TiO_2_ NPs were analyzed in R-RPMI 1640 medium to understand their behavior in cell cultures. The analyses in distilled water were performed to observe possible changes in nanoparticles behavior in the stock over time. UV/Vis spectra, hydrodynamic size, zeta potential, and TEM were used for the NPs characterization. TiO_2_ NPs dispersed in distilled water showed an aggregation behavior at the greatest concentration (100 µg/mL), and the size remained approximately the same between 2 and 24 h. The zeta potential was also stable at all TiO_2_ NPs concentrations (Table 1). However, different NPs behavior was observed when dispersed in the R-RPMI 1640 culture medium. Between 2 and 6 h of incubation, changes were not observed in the size distribution, while zeta potential indicated instability (Table 1). Between 6 and 24 h, we observed a great increase in particle size distribution in comparison with previous intervals. These changes indicate that NPs were dispersed in R-RPMI 1640 medium, and they started to precipitate only after 6 h of incubation. Magdolenova and colleagues assessed the relationship between the cytotoxic effects and the dispersion of TiO_2_ NPs [25]. They showed that tested cell culture medium types did not influence TiO_2_ NPs dispersion. However, they observed that different dispersion protocols and the use of serum in stock solution affected nanoparticles aggregation and size distribution. Accordingly, Ji et al. showed the improvement in TiO_2_ NPs dispersion upon addition of bovine serum albumin (BSA), although the dispersion also depended on cell culture media phosphate concentration [26]. TiO_2_ NPs tended to aggregate in R-RPMI 1640 medium (Table 1), which may be related to the low FBS concentration or the effect of phosphate ions in the cell culture medium.

By TEM, the aggregation of 500 µg/mL TiO_2_ NPs was also observed in distilled water (Figure 1). Therefore, it was not possible to determine the nanoparticles’ size. UV/Vis spectra were similar in exposed and control samples in both distilled water and R-RPMI 1640, as well as during the experiment, indicating that NPs properties did not change. Previously, the addition of HEPES and FBS into RPMI-1640 medium led to NPs re-dispersion [14]. 

Scanning electron microscopy showed the NPs cluster in contact with the cell membranes (Figure 2). EDS spectra showed TiO_2_ NPs that are present on cells at the 100 µg/mL concentration (Figure 3), but not at the lesser concentration (10 µg/mL). This may be because of the EDS microanalysis detection limit. TiO_2_ NPs are internalized by *E. fetida* coelomocytes. Bigorgne et al. determined their presence in the cell cytoplasm, but not in the nucleus or mitochondria [14]. However, we were unable to detect TiO_2_ NPs inside coelomocytes. This could be because TiO_2_ NPs aggregates are large. Earthworm coelomocytes are probably unable to engulf large NP clusters via phagocytosis and/or endocytosis, the most probable routes of TiO_2_ NPs entry into coelomocytes [1,14]. Phagocytic cells are potentially the most affected because they engulf NPs. Coelomocyte viability was not affected by exposure to 1, 10, and 100 µg/mL TiO_2_ NPs for 2, 6, and 24 h. Similar results were observed in *E. fetida* coelomocytes exposed to TiO_2_ NP [14]. Nanoparticles often trigger reactive oxygen species (ROS) production in cells, resulting in biomolecule oxidative damage [27,28]. We did not detect any statistically significant differences in ROS production in TiO_2_ NPs-exposed cells in comparison with control cells (Figure 4). Cells exposed to other nanoparticles, such as Ag NPs, nZVI NPs or ZnO NPs release significantly greater ROS amounts. Contrary to TiO_2_ NPs, ROS production could be elicited by the metal ions released from these nanoparticles [11,29,30]. 

We evaluated the apoptotic process in cells treated with TiO_2_ NPs, and did not detect any significant differences between exposed and control coelomocytes (Figure 5 and Figure 6). Late apoptosis was similar in GA and HA, with the greatest difference observed after 24 h of incubation. HA population exerted relatively greater early apoptosis than GA. Excess ROS production led to decreased cell viability and apoptosis [31,32]. Homa et al. suggested that coelomocytes are susceptible to bacterial or fungal products that may induce programmed cell death [31]. TiO_2_ NPs did not increase ROS production, and simultaneously, apoptosis was not increased as compared to control cells (Figure 4, Figure 5 and Figure 6). We suggest that TiO_2_ NPs do not affect ROS production, and thus do not trigger the apoptotic pathway in amoebocyte subpopulations (HA and GA). 

Amoebocytes are earthworm immune effector cells with the ability to phagocytose. At the phagocytic activity level, control cells and cells exposed to TiO_2_ NPs (1, 10, and 100 µg/mL) did not show any statistically significant changes (Figure 7). The results are in accordance with Bigorgne et al., who reported that there were no phagocytic activity changes in coelomocytes exposed to 1, 5, 10, and 25 µg/mL of TiO_2_ NPs, although TEM images demonstrated that TiO_2_ NPs were engulfed by the coelomocytes [14]. Thus, we can confirm that phagocytic activity is not compromised due to TiO_2_ NPs exposure.

ROS production initiates harmful radical chain reactions on cellular macromolecules, including DNA mutation, protein denaturation, and lipid peroxidation. At the lipid peroxidation level, MDA production was similar in both exposed and control cells. MDA is a subproduct derived from the reaction of free radical species with fatty acids [33]. We did not observe elevated lipid peroxidation (Figure 8). Ayala et al. explained that MDA is more stable and has a greater lifespan than ROS, and therefore it is more toxic [33]. Therefore, it could be a better biomarker for cellular oxidative stress detection. Excess ROS leads to MDA production [34]. Two oxidative stress markers, ROS and MDA, were produced at similar levels in control cells and TiO_2_ NPs-exposed cells (Figure 4 and Figure 8). The same results were also observed after THP1 human cells and sea urchin cells were exposed to TiO_2_ NPs [20,35]. UVA light could also enhance ROS production and increase toxicity several fold [36]. However, in this instance, the cells were mimicking the environmental conditions in the soil ecosystem, where UVA light was not present.

Significant differences between exposed and control cells were not detected regarding DNA damage. The alkaline comet assay results showed that there is no significant DNA damage in coelomocytes exposed to 1, 10, and 100 µg/mL of TiO_2_ NPs for 2, 6, and 24 h (Figure 9). A relationship between ROS, MDA, and DNA damage has been suggested. As mentioned previously, ROS may induce MDA production, which, in turn, affects nucleosides and results in DNA damage [33,34]. This mechanism has been described in coelomocytes exposed to pollutants, antibiotics, or pathogens [34]. Reeves et al. showed that GFSk-S1 cells (primary cell line from goldfish skin) exposed to different doses of TiO_2_ NPs (1, 10, and 100 µg/mL) could result in slight DNA damage, whereas co-exposure with UVA caused a significant increase in toxicity [36]. In vitro analysis described in this study did not reveal substantial changes in cellular physiologic activities, but the long-term exposure experiments can reveal different findings [37]. Zhu et al. described transcriptomic and metabolomic changes in earthworms as a global response to TiO_2_ NPs exposure that cannot be observed by conventional toxicity endpoints [38].

Treatment of coelomocytes with TiO_2_ NPs induced slight changes in the mRNA levels of distinct molecules. Metallothioneins are proteins protecting against metal-induced oxidative stress [9]. Metallothionein was upregulated in coelomocytes exposed to 10 µg/mL TiO_2_ NPs for 6 h, respectively (Table 2). This is in agreement with Bigorgne et al., who immuno-stimulated coelomocytes with lipopolysaccharides (LPS) (500 ng/mL) for 5 h prior to TiO_2_ NPs addition. After 12 h of incubation with 10 and 25 µg/mL TiO_2_ NPs, metallothioneins were upregulated [14]. We determined that even 1 µg/mL TiO_2_ NPs concentration upregulated metallothionein expression during the whole experiment (Table 2). Interestingly, the highest upregulation was detected in coelomocytes incubated with 1 µg/mL TiO_2_ NPs already after 2 h. Further, the induction of metallothionein expression in cells exposed to 10 µg/mL TiO_2_ NPs started at 6 h, and afterward decreased after 24 h (Table 2). Bigorgne et al. similarly showed increased metallothioneins expression after 12 h of incubation, with a subsequent decrease after 24 h [14].

Further, the antioxidant enzymes were not affected, except for Mn-SOD, which was downregulated after 6 h of coelomocyte exposure to 10 µg/mL TiO_2_ NPs (Table 2). Mn-SOD is a mitochondrial protein that protects cells against oxidative stress [39]. It seems that macrophages (RAW 264.7 cell line) and coelomocytes can engulf TiO_2_ NPs. These nanoparticles affect mitochondria even if they are not located inside the mitochondria [12,14]. Moreover, TiO_2_ NPs decreased ATP production in the macrophage RAW 264.7 cell line [12]. Thus, engulfed TiO_2_ NPs could target mitochondria and cause mitochondrial malfunction [12]. Mn-SOD downregulation and loss in mitochondrial oxidative phosphorylation function was also reported in primary rat hepatocytes [40]. 

Elevated levels of the antimicrobial proteins fetidin/lysenin and lumbricin were detected in cells exposed to 10 µg/mL TiO_2_ NPs for 6 h (Table 2). Similarly, Bigorgne et al. observed that fetidin was upregulated in cells exposed to 10 µg/mL TiO_2_ NPs after 12 h [14]. As previously described in related earthworm species *E. fetida*, lysenin regulation is changed rapidly by environmental stressors and it is suggested as an early biomarker of stress [41]. However, we cannot exclude that the increase in antimicrobial protein mRNA levels could be caused by used TiO_2_ NPs that were not LPS-free.

Referring to the signal transduction molecules, PKC I was strongly downregulated after coelomocyte exposure to 10 µg/mL TiO_2_ NPs for 6 and 24 h (Table 2). PKC I is important in cellular homeostasis and is involved in the cell proliferation signaling cascade [42,43]. This downregulation could suggest a coelomocyte homeostasis destabilization upon TiO_2_ NPs exposure. Another signal transduction molecule, MEKK, was upregulated in coelomocytes exposed to 1 µg/mL TiO_2_ NPs for 24 h and in coelomocytes exposed to 10 µg/mL TiO_2_ NPs for 6 h (Table 2). This molecule is involved in the MAPK cascade participating in many cellular processes, besides others in stress signaling [9,43]. Generally, coelomocyte exposure to TiO_2_ NPs results in slight changes in the mRNA levels of various molecules, however, these changes seem not to be significant enough to affect the observed cellular functions.

## 5. Conclusions

Coelomocytes exposed to TiO_2_ NPs (1, 10, and 100 µg/mL) did not show any impaired cellular responses as compared to control cells. The oxidative stress pathway and phagocytic activity were not affected as well. Nanoparticles do not cause greater DNA damage in treated cells than in non-treated cells. We also detected some gene expression alterations involved in metal detoxification, oxidative stress, defense reactions, and signal transduction. However, these changes do not seem to affect the observed cellular functions. In summary, we did not determine any detrimental effects of TiO_2_ NPs on *E. andrei* coelomocytes.

## Figures and Tables

**Figure 1 nanomaterials-11-00250-f001:**
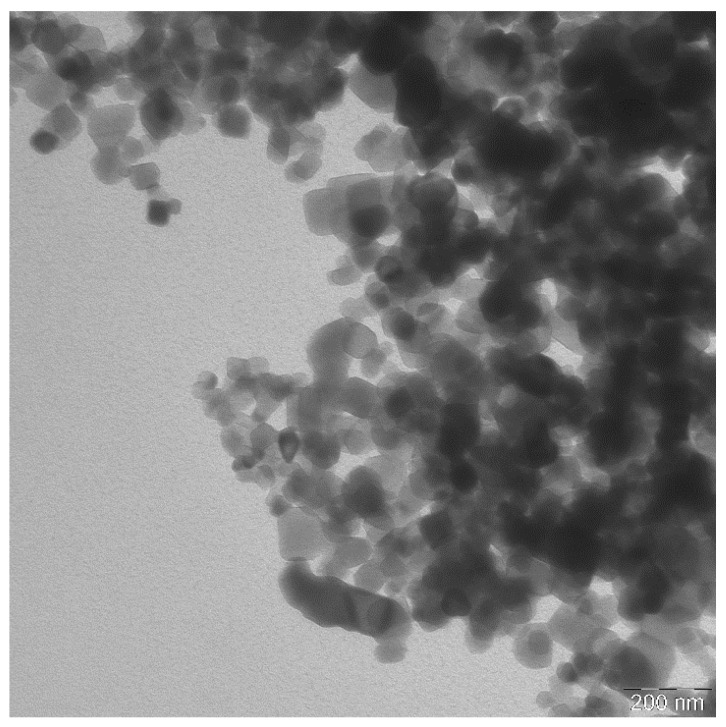
Transmission electron microscopy of 500 µg/mL TiO_2_ NPs clustered in distilled water. The scale bar represents 200 nm.

**Figure 2 nanomaterials-11-00250-f002:**
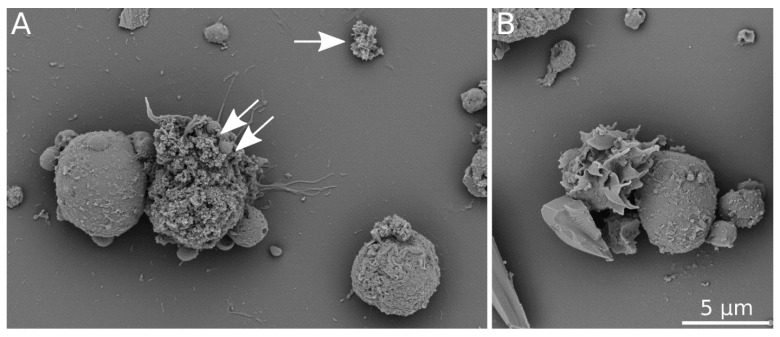
Scanning electron microscopy of coelomocytes. (**A**) cells exposed to 100 μg/mL TiO_2_ NPs for 2 h; (**B**) control cells cultured in the medium. Images recorded with B + C segments of a CBS detector at 3 kV. White arrow indicates a TiO_2_ NPs cluster on sample support. Clusters of the same morphology can be seen on the cell surface (white double arrow). The scale bar represents 5 μm.

**Figure 3 nanomaterials-11-00250-f003:**
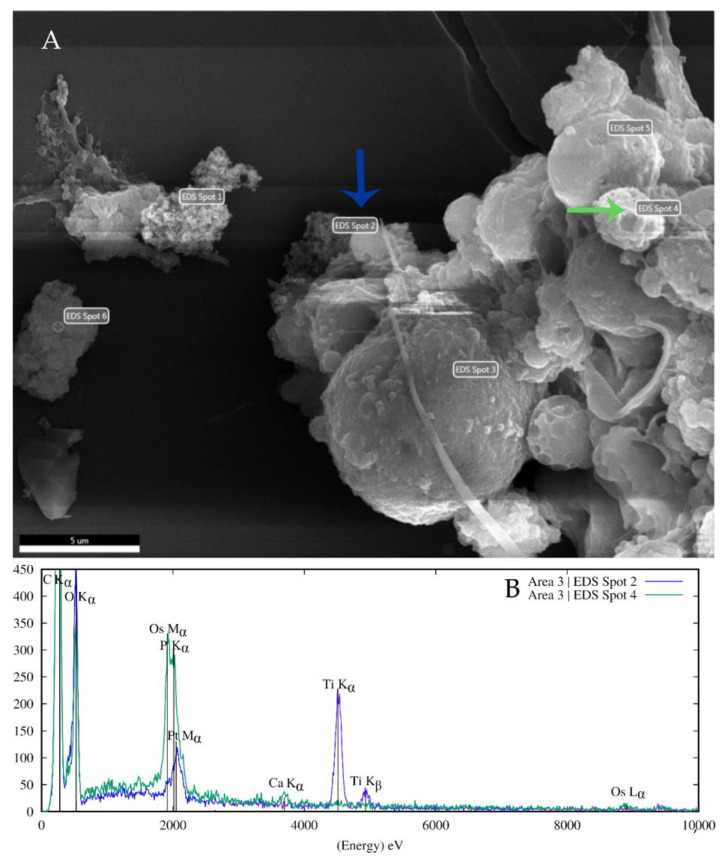
EDS microanalysis of coelomocytes incubated with 100 µg/mL TiO_2_ NPs. (**A**) An image showing the area of interest taken with EDX TEAM software at 15 kV using a SED detector. The spectra collection places are marked with EDS labels. Increased charging effects caused by the non-conductive nature of Thermanox coverslips used for sample preparation deteriorated image quality. (**B**) EDS microanalysis confirmed Ti in NPs clusters found on the cell surface (e.g., EDS Spot 2 label) and also in the cluster labeled EDS spot 1. Blue arrow indicates TiO_2_ NPs cluster (EDS Spot 2 label), green arrow points to the cell surface without NPs clusters (EDS Spot 4 label). Corresponding EDS spectra in matching colors are shown in B. The scale bar represents 5 µm.

**Figure 4 nanomaterials-11-00250-f004:**
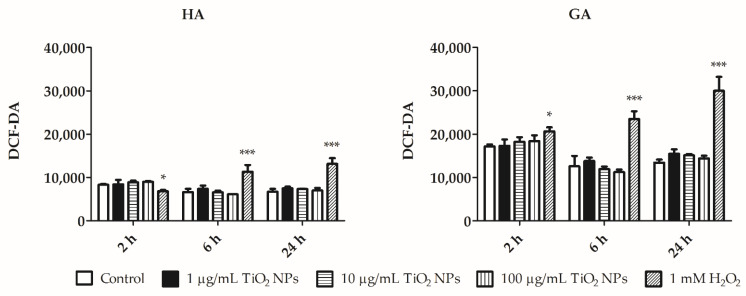
ROS production by hyaline (HA) and granular (GA). ROS production was measured in HA and GA after incubation with 1, 10, and 100 µg/mL TiO_2_ NPs for 2, 6, and 24 h using a cell-permeant tracer 2′,7′-dichlorofluorescein diacetate (DCF-DA). Coelomocytes were also exposed to 1 mM H_2_O_2_ (positive control) for 30 min. The results are shown as the mean of fluorescence intensity (DCF-DA) ± SEM of three independent experiments with 3 replicates in each. *** *p* < 0.001, and * *p* < 0.05 according to two-way ANOVA and Bonferroni post-test.

**Figure 5 nanomaterials-11-00250-f005:**
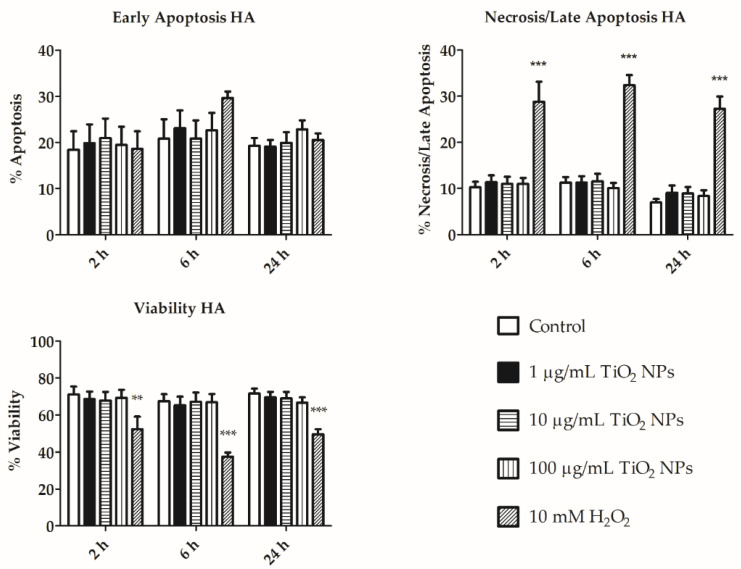
Early and late apoptosis, viability and necrosis of hyaline amoebocytes (HA). Early and late apoptosis, viability and necrosis of HA of non-treated cells, cells exposed to 1, 10, and 100 µg/mL TiO_2_ NPs after 2, 6, and 24 h. 10 mM H_2_O_2_ was used as positive control for 30 min exposure. The results are shown as mean (%) ± SEM of three independent experiments with 3 replicates in each. *** *p* < 0.001, and ** *p* < 0.01 according to two-way ANOVA and Bonferroni post-test.

**Figure 6 nanomaterials-11-00250-f006:**
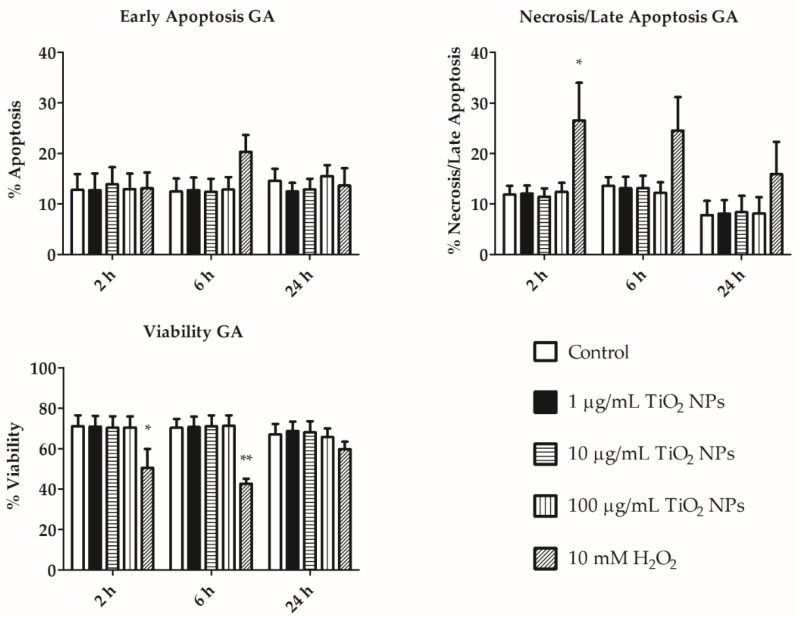
Early and late apoptosis, viability and necrosis of granular amoebocytes (GA). Early and Late apoptosis, viability and necrosis of GA of non-treated cells, cells exposed to 1, 10, and 100 µg/mL TiO_2_ NPs after 2, 6, and 24 h. 10 mM H_2_O_2_ was used as positive control for 30 min exposure. The results are shown as mean (%) ± SEM of three independent experiments with 3 replicates in each. ** *p* < 0.01, and * *p* < 0.05 according to two-way ANOVA and Bonferroni post-test.

**Figure 7 nanomaterials-11-00250-f007:**
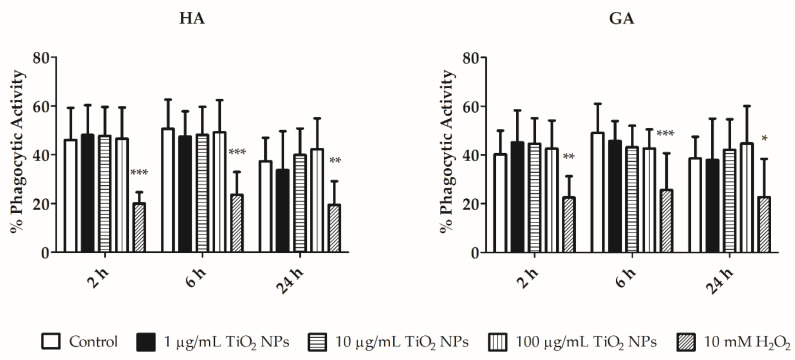
Phagocytic activity of HA and GA. Phagocytic activity was measured after incubation with TiO_2_ NPs (1, 10, and 100 µg/mL) for 2, 6, and 24 h. Coelomocytes were also exposed to 10 mM H_2_O_2_ (positive control) for 30 min. Results are represented as the mean ± SEM of three independent experiments with 3 replicates in each. *** *p* < 0.001, ** *p* < 0.01, and * *p* < 0.05 according to two-way ANOVA and Bonferroni post-test.

**Figure 8 nanomaterials-11-00250-f008:**
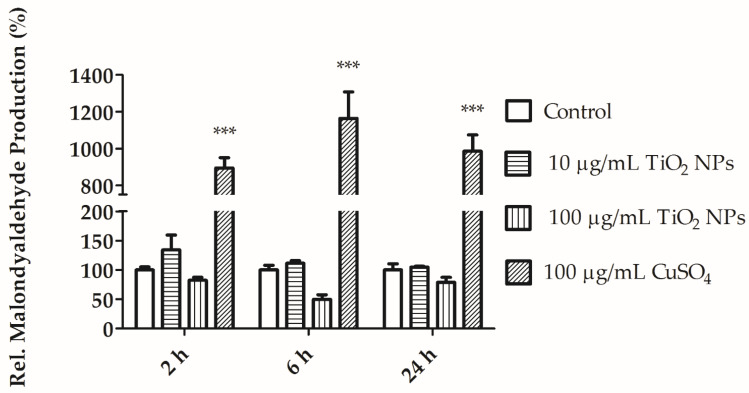
Relative malondialdehyde (MDA) production in coelomocytes exposed to 10, 100 µg/mL TiO_2_ NPs and positive control (100 µg/mL CuSO_4_) for 2, 6, and 24 h. Values are expressed as mean (%) ± SEM of three independent experiments each with three replicates. *** *p* < 0.001 according to two-way ANOVA and Bonferroni post-test.

**Figure 9 nanomaterials-11-00250-f009:**
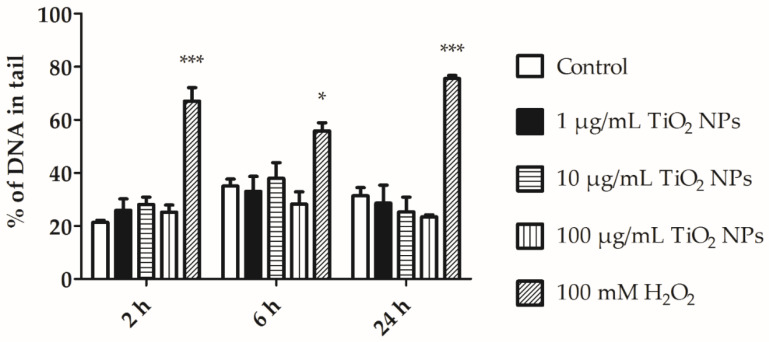
DNA damage in coelomocytes after their exposure to 1, 10, and 100 µg/mL TiO_2_ NPs for 2, 6, and 24 h. Coelomocytes were also exposed to 100 mM H_2_O_2_ (positive control) for 30 min. Values are expressed as the mean of DNA content in tail (%) ± SEM of three experiment with three replicates. *** *p* < 0.001 and * *p* < 0.05 according to two-way ANOVA and Bonferroni post-test.

**Table 1 nanomaterials-11-00250-t001:** Characterization of 100 µg/mL TiO_2_ nanoparticles (NPs) suspension in milliQ water and R-RPMI 1640 medium.

	UV/Vis (nm) ^a^			Z-Avg. (nm) ^b^			ζ (mV) ^c^		
	2 h	6 h	24 h	2 h	6 h	24 h	2 h	6 h	24 h
**Distilled water**	300–370	300–370	300–370	581 ± 23.30	570 ± 2.75	480 ± 64.3	−26.8 ± 2.99	−31.7 ± 0.921	−32.9 ± 2.59
**R-RPMI 1640 medium**	320–380	320–380	320–380	31.34 ± 1.55	35.5 ± 3.94	597 ± 447	−16.9 ± 0.60	−7.87 ± 0.631	−5.94 ± 0.45

(a) ultraviolet-visible = UV/Vis spectra absorbance (nm), (b) Z-Avg = Hydrodynamic size determined by multi-angle dynamic light scattering (MADLS), and (c) ζ = zeta potential values are expressed as mean of 3 measurements ± SD.

**Table 2 nanomaterials-11-00250-t002:** The mRNA levels of distinct molecules in coelomocytes exposed to 1 and 10 µg/mL TiO_2_ NPs.

Function	Gene	TiO_2_ NPs (µg/mL)	Normalized Gene Expression		
			2 h	6 h	24 h
**Metal detoxification**	Metallothionein	1	**5.16 ± 1.73 ****	**2.00 ± 0.32 ***	**2.71 ± 0.20 ***
		10	1.11 ± 0.2	**1.97 ± 0.22 ****	1.00 ± 0.25
**Heavy metal detoxification**	Phytochelatin	1	1.38 ± 0.09	1.02 ± 0.04	1.18 ± 0.08
		10	1.00 ± 0.02	0.82 ± 0.02	0.80 ± 0.01
**Oxidative stress**	Mn-SOD	1	1.47 ± 0.12	0.85 ± 0.19	0.58 ± 0.05
		10	0.93 ± 0.09	**0.53 ± 0.01 ***	0.72 ± 0.01
	CuZn-SOD	1	0.68 ± 0.05	0.84 ± 0.22	0.98 ± 0.04
		10	0.96 ± 0.07	0.71 ± 0.04	0.87 ± 0.01
	Catalase	1	1.41 ± 0.19	0.87 ± 0.03	0.66 ± 0.03
		10	1.04 ± 0.13	0.71 ± 0.02	0.8 ± 0.2
**Immunity**	EMAP II	1	0.90 ± 0.07	0.94 ± 0.1	0.86 ± 0.02
		10	0.84 ± 0.09	1.21 ± 0.01	1.33 ± 0.20
	Fetidin/lysenin	1	0.64 ± 0.08	0.62 ± 0.13	0.70 ± 0.04
		10	0.65 ± 0.05	**2.20 ± 0.2 ****	0.81 ± 0.19
	Lumbricin	1	1.33 ± 0.05	0.75 ± 0.10	1.84 ± 0.02
		10	0.84 ± 0.10	**2.10 ± 0.43 ***	1.92 ± 0.55
**Signal Transduction**	MEKK I	1	1.40 ± 0.19	1.47 ± 0.44	**1.73 ± 0.04 ***
		10	1.00 ± 0.15	**1.96 ± 0.11 ***	1.33 ± 0.03
	PKC I	1	1.52 ± 0.30	1.08 ± 0.19	1.43 ± 0.06
		10	1.10 ± 0.16	**0.33 ± 0.04 ****	**0.58 ± 0.11 ***

Values were normalized to two reference molecules (RPL13 and RPL17). Fold changes (±SEM) in mRNA levels in TiO_2_ NPs exposed coelomocytes are relative to the mRNA levels in control cells. Two-way ANOVA and Bonferroni post-test were performed to evaluate data significance (* *p* < 0.05, ** *p* < 0.01). mRNA levels quantification was performed by three independent experiments with duplicates per each treatment and time interval. Mn-SOD: manganese superoxide dismutase; CuZN-SOD: copper-zinc-superoxide dismutase; EMAP II: endothelial monocyte-activating polypeptide-II; MEKK I: MEK kinase I; PKC I: protein kinase C I.

## Data Availability

Data is contained within the article or Appendix A.

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
