# Peer review of "In Vitro Interactions of TiO2 Nanoparticles with Earthworm Coelomocytes: Immunotoxicity Assessment"

_nanomaterials, 2021, doi:10.3390/nano11010250_

Round 1
Reviewer 1 Report
Reviewer opinion about nanomaterials-1067093
The aforementioned MS by Navarro Pacheco et al., have potential scientific merit but it needs some revision.
Major points:
- Materials & Methods: page 2, line 76-77. Please be more specific for the composition of the extrusion buffer, did it contain ethanol?
- M&M, page 4, line 156-158 What is the applied concentration of DCF ?
- Page 4, line 162. What type of AnnexinV conjugate has been used? According to the supplementary it is an APC conjugate. However it would be more straightforward if it appears in the main text as well.
- Page 5, line 232, During the NP physical characterization RPMI was complemented with FBS? If, yes at what percentage?
- page 6, in table 1, the hydrodynamic size (480-581 nm) represent aggregated TiO NPs in the case of distilled water at different time points. What can be the explanation for that? It would be good if TEM image provided for the exact exposure conditions as well.
- page 7, Figure 2, please provide a control SEM image, where coelomocytes were not exposed to TiO NPs.
- page 8, Figure 3, please provide EDX spectra for the control samples as well.
- page 9. For the flow cytometry analysis can you provide an SSC analysis in the case of different concentration of TiO NPs? It would prove that NPs are not internalized.
- page 22, line 386-400, Annexin V staining: please provide evidence how can you distinguish late apoptotic vs necrotic cells. Several publication claim that Annexin V/PI positive cells are actually can be both late apopotic and necrotic cells as well.
Minor point
- Page 12, Figure 8. Figure legends have some typo errors regarding to the applied concentrations (ug vs. mg).
Author Response
revision.
Major points:
- Materials & Methods: page 2, line 76-77. Please be more specific for the composition of the extrusion buffer, did it contain ethanol?
The extrusion buffer did not contain ethanol. We have the best experiences with applying extrusion buffer to worms, which contains guaiaocal glyceryl ether (expectorant), EDTA and diluted PBS (LBSS).
- M&M, page 4, line 156-158 What is the applied concentration of DCF ?
We apologize for the unclear information. We used the stock solution of 20.6 mM CDF-DA. Then, we diluted it 1000times. The final concentration of DCF-DA was 20.6 uM. We corrected it in M&M section.
- Page 4, line 162. What type of AnnexinV conjugate has been used? According to the supplementary it is an APC conjugate. However it would be more straightforward if it appears in the main text as well.
We apologize for missing information. We used Alexa Fluor 647-Annexin V conjugate. We added this information to the materials and methods section.
- Page 5, line 232, During the NP physical characterization RPMI was complemented with FBS? If, yes at what percentage?
During all experiments with RPMI, RPMI medium supplemented with 5% heat-inactivated fetal bovine serum was used, as is written in the section 2.1 of the material and methods.
- page 6, in table 1, the hydrodynamic size (480-581 nm) represent aggregated TiO NPs in the case of distilled water at different time points. What can be the explanation for that? It would be good if TEM image provided for the exact exposure conditions as well.
The relatively wide distribution in the hydrodynamic size of TiO2 NPs (aggregates) over time was very probably caused by the heterogeneity of the commercial stock suspension together with fast sedimentation of suspended NPs. Both factors could affect manipulation with NPs during pretreatment (dilution, etc.) and result in different concentrations and thus different aggregation kinetics. The standard deviation reaching values of ±64.3 nm reflects the variability of the presented results and supports the previous hypothesis. Moreover, the decreasing trend in the hydrodynamic size of the aggregate over time is not in accordance with other parameters and therefore we believe that the size of the NP clusters will be comparable within 2–24 h.
We had a severe problem with the aggregation/agglomeration of TiO2 nanoparticles, please see the figure. We have tried pure carbon and formvar/carbon support films, glow-discharge activated or not. In all cases, the aggregation/agglomeration of NPs was the problem. The accompanying image documents the big NPs clusters found in the freshly prepared TEM samples without any incubation. The sample preparation procedure: The commercial stock NPs was diluted to 100 μg/ml or 500 μg/ml in ddH2O or 100 μg/ml in a culturing medium. Within several minutes, the samples were prepared. Before applying the NPs mixture onto the TEM grid, the Eppendorf tube with the NPs mixture was briefly vortexed.
Because we have observed an enormous aggregation/agglomeration of TiO2 NPs in those samples, we did not perform the experiment you have suggested. It would be pretty nice to have such data from TEM, but we cannot get the TEM samples without NPs aggregation/agglomeration in our setup. In our experiences with other types of NPs, for gold and silver NPs, we could minimize the aggregation of them. However, whenever we worked with NPs based on (metal)oxides (CuO, Fe2O3/Fe3O4, SiO2, TiO2), the NPs aggregation/agglomeration was always a problem.
- page 7, Figure 2, please provide a control SEM image, where coelomocytes were not exposed to TiO NPs.
We prepared a new figure 2 containing a control SEM image as well and we added it to the manuscript.
- page 8, Figure 3, please provide EDX spectra for the control samples as well.
We provided EDS spectra for control samples as well. They were added to the manuscript as new Figure S2.
- page 9. For the flow cytometry analysis can you provide an SSC analysis in the case of different concentrations of TiO NPs? It would prove that NPs are not internalized.
Unfortunately, our Instrument used for flow cytometry analysis is not so sensitive to distinguished the changes in light refraction (SSC) caused by different concentrations of NPs. Nevertheless, the plots of cell distribution for all NPs concentrations are enclosed (new Figure S3). The samples where TiO2 NPs were used without cells showed the gate where we could find non-internalized NPs (new Figure S1).
- page 22, line 386-400, Annexin V staining: please provide evidence how can you distinguish late apoptotic vs necrotic cells. Several publication claim that Annexin V/PI positive cells are actually can be both late apopotic and necrotic cells as well.
We agree with the reviewer that distinguishing these two populations is difficult. Therefore, we decided to joint Necrosis and late apoptosis signals. New graphs on this data were performed (figures 5 and 6).
Minor point
Page 12, Figure 8. Figure legends have some typo errors regarding to the applied concentrations (ug vs. mg).
We thank the reviewer for this comment. We corrected the typos in figure 8.

Reviewer 2 Report
There are many studies concerning titanium dioxide (TiO2) nanoparticles (NP) toxicity but the presented work is interesting and shows the influence of the factor on invertebrates cells in a wide range of results. The work is done very carefully and contains many interesting results.
I have a few comments.
In materials and methods, in paragraph 2.4 Flow Cytometry Assays, the authors did not provide information about annexin V conjugation. I found information that it was APC only on the figures description. I have also a question about the volume of annexin V. Normally 5 ul or less is used, why the authors give 30 ul per sample?
One wondering is the question of aggregates/agglomerates of nanoparticles, whether it is possible to reduce their formation in order to standardize the study of the nano impact? Perhaps it would be a good move to prevent the formation of agglometers cluster of nano, reducing nanoparticle agglomeration e.g. membrane filter or using a sonicator? This is not a objection relating to this work, but the interesting question is the effect of different particle size of nano cluster.
The results indicate that the short times did not affect the activity of the coelomocytes, which is not definite what would be the case with a longer exposure of the organism in in vivo treatment (days or weeks) maybe it would be good discussed this.
Supplementary data:
In the supplementary file some of the words are connected, probably it results from my Office, but in several places I marked the potential problem e.g.:
page 5, line 71-73, Figure S5. Illustrative figure of phagocytic activity of GAafter 2 hours. Phagocytic activity of cells withoutFluoresbrite beads and PI, non-treated cells,cells treated with 10 mM H2O2, cells treated with 100, 10, and 1 µg/mL TiO2 NPs.
page 5, line 99: 1µmmicrosphere and cellsafter
Figure S2. figure 2 is very detailed, it might be good to simplify it by using common axes (FITC-A and % of Max) and use one common abbreviation for GA and HA with the appropriate panel.
On the Figure S3 and Figure S4 it is lack of plot with only Annexin V. It would be good to show a complete set of figures e.g. A) without staining, B) only PI, C) annexin V alone, and next three plots D) control, E) TiO2 NPs, and F) H2O2.
References:
Reference number 30 and 32, the same work cited twice. Please check if the numbering is correct after the change.
Standardize the method of citation, e.g. 20, 40. abbreviations or full name of journal
page 19, line 649: Eisenia andrei lack italic

Author Response
Reviewer 2:
There are many studies concerning titanium dioxide (TiO2) nanoparticles (NP) toxicity but the presented work is interesting and shows the influence of the factor on invertebrates cells in a wide range of results. The work is done very carefully and contains many interesting results.
I have a few comments.
- In materials and methods, in paragraph 2.4 Flow Cytometry Assays, the authors did not provide information about annexin V conjugation. I found information that it was APC only on the figures description. I have also a question about the volume of annexin V. Normally 5 ul or less is used, why the authors give 30 ul per sample?
We apologize for missing information. We used Alexa Fluor 647- Annexin V conjugate. We used different volumes of Annexin dilutions according to the volume of cells. However, we used 5 ul of stock solution per sample. We added this information to the materials and methods section.
- One wondering is the question of aggregates/agglomerates of nanoparticles, whether it is possible to reduce their formation in order to standardize the study of the nano impact? Perhaps it would be a good move to prevent the formation of agglometers cluster of nano, reducing nanoparticle agglomeration e.g. membrane filter or using a sonicator? This is not a objection relating to this work, but the interesting question is the effect of different particle size of nano cluster.
To prevent the agglomeration of studied NPs is the objective of all researchers in relation to studying the real effect of nanoscale materials. However, the aggregation process depends on many factors (e.g. ionic strength, pH, the composition of exposition media or NPs surface charge) and it is not always possible to avoid this, often natural, environment-NPs interaction. We agree with the reviewer that a membrane filter or sonicator could reduce the aggregation of nanoparticles; however, we strongly believe that even the information about the possible adverse effects of aggregated NPs with proper characterization will provide new information in the field of NPs risk assessment. Moreover, to provide comparable results within the EU Project PANDORA, the same protocol of NPs dispersion (pretreatment) was used.
- The results indicate that the short times did not affect the activity of the coelomocytes, which is not definite what would be the case with a longer exposure of the organism in in vivo treatment (days or weeks) maybe it would be good discussed this.
We agree with the reviewer that the results of in vitro experiments, which can be performed for only a limited time due to the short viability of earthworm coelomocytes, can differ from the long-term experiments in vivo. For example, Hu et al. showed harmful effects of TiO2 NPs presented at high levels in soil on E. fetida. Similarly, Zhu described transcriptomic and metabolomic changes of earthworms as global response to TiO2 NPs exposure. We added this point to the discussion.
Hu et al. 2010. Toxicological effects of TiO2 and ZnO nanoparticles in soil on earthworm Eisenia fetida. Soil Biology and Biochemistry.
Zhu te al. 2020. Integration of transcriptomics and metabolomics reveals the responses of earthworms to the long-term exposure of TiO2 nanoparticles in soil. Science of Total Environment.
- Supplementary data: In the supplementary file some of the words are connected, probably it results from my Office, but in several places I marked the potential problem e.g.: page 5, line 71-73, Figure S5. Illustrative figure of phagocytic activity of GAafter 2 hours. Phagocytic activity of cells withoutFluoresbrite beads and PI, non-treated cells,cells treated with 10 mM H2O2, cells treated with 100, 10, and 1 µg/mL TiO2 page 5, line 99: 1µmmicrosphere and cellsafter
We checked the supplementary file and we don´t see a described problem with missing spaces in the text. Probably, it really can be a problem with the used Office.
- Figure S2. figure 2 is very detailed, it might be good to simplify it by using common axes (FITC-
We agree with the reviewer that the figure can be simplified. We modified the figure according to the reviewer's suggestion.
- On the Figure S3 and Figure S4 it is lack of plot with only Annexin V. It would be good to show a complete set of figures e.g. A) without staining, B) only PI, C) annexin V alone, and next three plots D) control, E) TiO2NPs, and F) H2O2.
- References:
Reference number 30 and 32, the same work cited twice. Please check if the numbering is correct after the change. Standardize the method of citation, e.g. 20, 40. abbreviations or full name of journal. page 19, line 649: Eisenia andrei lack italic
We thank the reviewer for this comment. We corrected the reference numbering.
We apologize for the discrepancies in reference formatting. We used reference manager Endnote and recommended style MDPI. However, from unknown reasons, the formating was not unified. We corrected all references.
